# Selective Detection of $Fe^{3+}$ by Nitrogen–Sulfur-Doped Carbon Dots Using Thiourea and Citric Acid

Heng Zhou [†], Ying Ren *,[†], Zheng Li, Weichun He and Zhengxin Li *

Engineering and Technology Research Center of Diamond Composite Materials of Henan, School of Materials Science and Engineering, Henan University of Technology, Zhengzhou 450001, China; zhouheng19980428@163.com (H.Z.); li_2863925189@163.com (Z.L.); weichun_he@haut.edu.cn (W.H.)
* Correspondence: ying_ren@haut.edu.cn (Y.R.); zhengxin_li@haut.edu.cn (Z.L.)
† These authors contributed equally to this work.

**Abstract:** The quantum yield and fluorescence properties of carbon dots are key issues for environmental detection. In this study, nitrogen–sulfur-doped carbon dots (N,S-CDs) were prepared hydrothermally by adding thiourea to provide the N source. By adjusting the ratio of citric acid (CA) to thiourea (N,S) and adding anhydrous ethanol, blue fluorescent doped carbon dots with a quantum yield of up to 53.80% were obtained. The particle morphology and crystalline organization of the N,S-CDs were analyzed using transmission electron microscopy (TEM) and X-ray diffraction (XRD). Fourier transform infrared (FTIR) spectroscopy was used to illuminate distinct functional units through the recording of typical vibration bands. The luminescence properties of the N,S-CDs were investigated using ultraviolet–visible (UV-vis) absorption spectroscopy and steady-state fluorescence spectroscopy (PL). In addition, the fluorescence stability of the N,S-CDs was studied in detail. The results showed that the functional groups of the N,S-CDs chelate $Fe^{3+}$ ions to quench the fluorescence of carbon dots. This shows that the N,S-CDs exhibit high selectivity for $Fe^{3+}$ ions. With the addition of $Fe^{3+}$ in the concentration of 0–100 μM, the fluorescence intensity of the N,S-CDs exhibited distinct and linear dependence upon the $Fe^{3+}$ concentration ($R^2$ = 0.9965), and the detection limit (D = 3σ/m) was measured as 0.2 μM. The excellent optical properties and $Fe^{3+}$ selectivity of the N,S-CDs provide a huge boost for application in the field of environmental monitoring.

**Keywords:** nitrogen and sulfur doping; fluorescence intensity; carbon dots; fluorescence probe; $Fe^{3+}$ detection





## 1. Introduction

Iron, one of the essential trace elements for all living organisms, has little toxicity to humans and animals, but an excess amount of $Fe^{3+}$ will lead to imbalanced human homeostasis and cause symptoms such as chronic poisoning, which seriously affect the healthy development of the human body [1–3]. In the water system, $Fe^{3+}$ accumulates in organisms through the food chain and cannot be directly degraded. In vivo, $Fe^{3+}$ can be transformed into a more toxic form or can directly interfere with metabolic processes [4–7]. When the concentration of $Fe^{3+}$ in water is 0.1–0.3 mg/L [8], it will affect the color, smell, and taste of the water. Moreover, some special industries have higher requirements for iron content in water, such as the textile [9], paper [10], brewing [11], and food industries [12]. Therefore, under the condition of strictly controlling the $Fe^{3+}$ content, researchers need to strictly detect $Fe^{3+}$ in water resources [4–7].

Therefore, how to quickly and accurately detect $Fe^{3+}$ in the environment has always attracted much attention [13]. At present, there are many detection methods for $Fe^{3+}$, including potential voltammetry [14], atomic absorption spectrometry [15], and colorimetric methods [16]. However, most of these testing methods require expensive tools or instruments, and the more complex testing processes also put forward higher technical requirements for sample processing and operation. In recent years, with the maturity

of fluorescent carbon dot detection technology, it has been widely applied to detect the concentration of $Fe^{3+}$ due to its advantages of simple and low-cost preparation [17,18], high sensitivity [19], instantaneous response, and the specific recognition of metal ion [20–22].

As new carbon nanomaterials, carbon dots have a large number of hydrophilic functional groups on the surface, which allow them to have good water solubility and excellent fluorescence properties. At present, there are two commonly used methods for the synthesis of carbon dots, namely microwave-based synthesis and hydrothermal synthesis. For microwave-assisted methods, the advantage is that they can quickly synthesize carbon dots. For example, Chen et al. [23] rapidly synthesized 12.45% QY carbon dots in 4 min utilizing a mixture of glutamic acid and ethylenediamine, which specifically responds to $Fe^{3+}$ in the linear range of 8–80 μM. It has been successfully applied to the detection of $Fe^{3+}$ in fungal cells due to its detection limit of 3.8 μM and its good water solubility. However, compared to the hydrothermal method, the linear range and detection limit can be further improved by using a simpler hydrothermal method to synthesize doped carbon dots, as reported in references [24–28].

Muhammad et al. [24] used cranberry beans to prepare carbon dots with a detection limit of 9.55 μM in the linear range of 30–600 μM $Fe^{3+}$, and $Fe^{3+}$ ion detection could be achieved in a short time. The carbon dots prepared by Zhu et al. presented a detection limit of 0.45 μM in the concentration range of 0–60 μM [25]. These carbon dots that were synthesized using a hydrothermal method have a better detection range and detection limit than the microwave method. On the basis of these studies, a great deal of effort is devoted to improving the detection limit and range by changing the raw materials. Pu et al. [26] used phenylalanine as a nitrogen source and a common carbon source to prepare carbon dots. The results showed that $Fe^{3+}$ combined with oxygen-containing functional groups outside of carbon dots formed static quenching, and $Fe^{3+}$ combined with the induced carbon dots performed selective quenching. The carbon dots have a linear relationship with relative strength in the detection range of 5–500 μM, and the detection limit is as low as 0.720 μM. However, on the original basis, by using an environmentally friendly carbon source, that is, changing the type of carbon source, carbon dots with a QY of 28.6% and a lower detection limit of 0.398 μM were successfully prepared [27]. Through the analysis of the quenching mechanism, it was found that the carbon dots have an obvious selective quenching effect on $Fe^{3+}$, and they also showed a good linear relationship in the range of 0–100 μM. Huang et al. reported the preparation of 8.6% QY copper nanocarbon dots (Cu-NCs) using glutathione as a stabilizer [28]. The nanocarbon dots were in the $Fe^{3+}$ concentration range of 1–100 μM, the fluorescence of Cu-NCs was linearly quenched, the detection limit was 0.3 μM, and the nanocarbon dots were available for the detection of $Fe^{3+}$ in real water samples. All of this proves that the detection of $Fe^{3+}$ by carbon dots has the characteristics of high efficiency, reliability, and high selectivity, which can make it widely used in the field of environmental monitoring. However, due to the high detection limit and low-fluorescence quantum yield, carbon dots still cannot be widely used as sensors for the detection of ions in food and production.

In this work, we aimed to synthesize nitrogen–sulfur-doped carbon dots (N,S-CDs) with high fluorescence intensity and fluorescence quantum yield using a hydorthermal method (as shown in Figure 1) and utilizing a mixed citric acid, thiourea, and ethanol solution that can provide heteroatomic sulfur, amino, and hydroxyl functional groups during the synthesis process. The synthesized carbon dots had good water solubility, photostability, a high quantum yield, a specific response to $Fe^{3+}$ ions, and high selectivity, which enable the carbon dots to be successfully applied in the field of environmental monitoring.

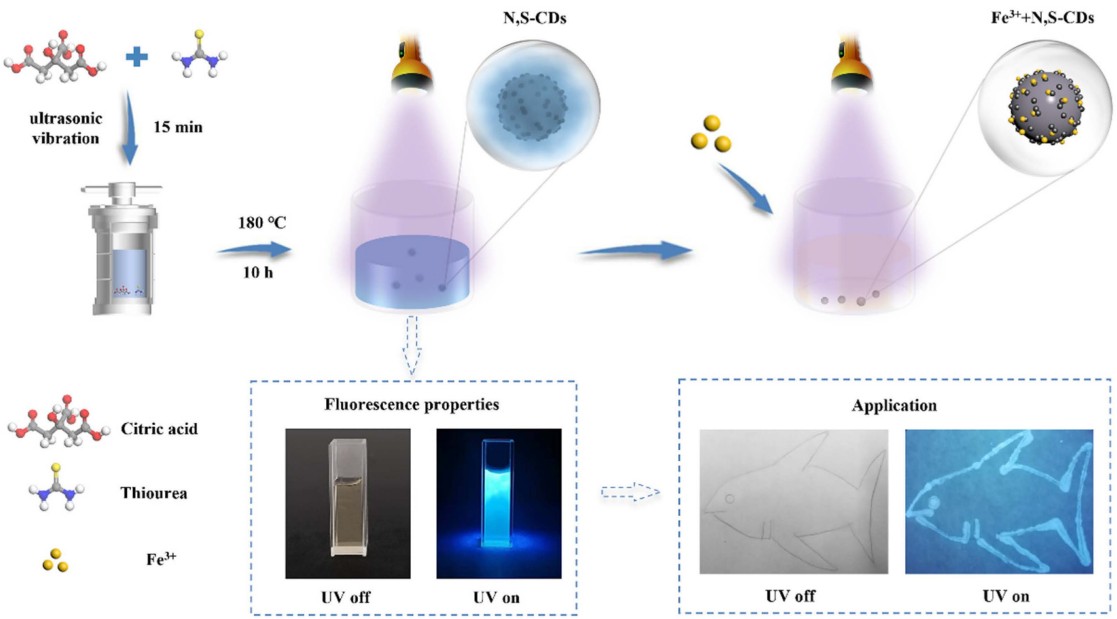

**Figure 1.** Schematic diagram of the preparation and application of N,S-CDs and their application in $Fe^{3+}$ sensing.

## 2. Experimental Section

### 2.1. Materials

All metal ion salts were analytically pure, including NaCl, KCl, $Ba(NO_3)_2$, $Bi(NO_3)_2$, $ZnCl_2$, $MgCl_2$, $Al(NO_3)_3$. The compounds $9H_2O$, $Co(NO_3)_2$, $AgNO_3$, $FeCl_3$, $Pb(NO_3)_2$, and $CuSO_4$ were all obtained from Aladdin. Citric acid monohydrate, thiourea, dialysis membrane (MWCO 1000 Da), and quinine sulfate (98%) were purchased from Aladdin (Shanghai, China). The water used in all experiments was deionized water.

### 2.2. Instruments

The fluorescence and UV spectra were determined by using a fluorescence spectrometer (Hitachi, Tokyo, Japan) and a UVprobe2.70 UV-Vis spectrophotometer (Shimadzu Corporation, Shimadzu, Japan). The FTIR spectra were determined by using an IS20 Fourier Transform Infrared Spectrometer (Thermo Fisher Scientific, Shanghai, China). A drop of an aqueous solution containing N,S-CDs was dropped into the KBr particles, and after the water had evaporated, the particles were used to measure the FTIR spectrum (Thermo Fisher Scientific, Shanghai, China). XRD pictures were obtained using a MINI-FLEX 600 benchtop X-ray diffractometer (Rigaku MiniFlex, Jiangsu, China). Transmission electron microscopy (TEM, Thermo Fisher, Shanghai, China) graphics were acquired on a FEI Tecnai G2 F20 high-resolution transmission electron microscope.

### 2.3. Preparation of N,S-CDs

Citric acid (CA) and thiourea (N,S) were first dissolved in a 10 mL absolute ethanol solution with the different substance ratios (2:1, 1:1, 1:2, 1:3, 1:4) and then were treated via ultrasonic dispersion for 10 min. The mixture was then added to the liner of a 50 mL Teflon reactor and heated at 180 °C for 12 h. The product obtained from the reaction was placed at room temperature for a period of time. The room temperature product was placed in a centrifuge (10,000 rpm, 30 min) to obtain the supernatant, and the filtrate containing the N,S-CDs was collected by filtration through a 0.22 μm syringe filter 2–3 times. Finally, the filtrate was put into a bag with a dialysis function (Mw = 1000 Da) for 3 days; the water was changed every 5 h to remove impurities; and the N,S-doped carbon dot products were acquired. In order to meet the needs of characterization, e.g., XRD, the sample needed to be prepared as a solid, and the method was as follows: the solution was placed in the

cold trap of a freeze dryer to freeze, and freeze-drying was performed to obtain a solid powder product.

### 2.4. Fluorescence Quantum Yield Determination

The fluorescence quantum yield was calculated using the reference method in [29,30]. Quinine sulfate was dissolved in 0.1 M sulfuric acid to obtain a quinine sulfate solution as a standard solution. The quantum yield was 0.54 at excitation wavelengths of 340 nm and 360 nm. The quantum yield of doped carbon dots is calculated as

$$QY_S = QY_R \times (F_S/F_R) \times (A_R/A_S) \times (N_S{}^2/N_R{}^2)$$

where QY indicates the fluorescence quantum yield, F indicates the fluorescence integral area at the excitation wavelength, A represents the absorbance (required to be kept at 0.05 or below), N indicates the refractivity of the solution, and the subscripts of S and R represent the carbon point solution and the quinine sulfate solution, respectively.

### 2.5. Detection of $Fe^{3+}$ Ions

$FeCl_3$ was used as the ion detection source, and differently weighted volumes were used. A 10 mL volumetric flask had 1 mM of $Fe^{3+}$ solution and 2 mL of carbon dot dialysate. Ultrapure water was used to dilute the solution to the calibration line. The final concentrations of diluted $Fe^{3+}$ were 0, 10, 20, 30, 40, 50, 60, 70, 80, 90, 100, 200, 300, 400, 500, and 600 μM. After shaking the volumetric flask up and down, the fluorescence intensity was measured at room temperature, and the process was repeated.

### 2.6. Selective Detection of Metal Ions

Under the same conditions as $Fe^{3+}$ ion detection, 12 kinds of 1 mM metal ion solutions ($Ag^+$, $Ba^{2+}$, $Bi^{2+}$, $Co^{2+}$, $Cu^{2+}$, $Fe^{3+}$, $Pb^{2+}$, $Na^+$, $Zn^{2+}$, $Al^{3+}$, $K^+$, $Mg^{2+}$) and 1 mL of carbon dot solution were mixed to prepare the test stock solution with a metal ion concentration of 500 μM. The metal ion selectivity experiments were performed at room temperature with 11 replicates for each set of experiments, and standard deviation error bars were calculated. In addition, after the determination of ion selectivity, stability testing was carried out. For example, under the premise of the same concentration of N,S-CDs, the addition of different concentrations of NaCl solution and pH was conducted to explore the stability of the N,S-CDs. Relative ratio ($F/F_0$) of the fluorescence emission intensities of the N,S-CDs in the presence and absence of metal ions was used to indirectly indicate the stability of the N,S-CDs.

### 2.7. Real Samples Measurement

To evaluate the feasibility of $Fe^{3+}$ carbon spot detection in real samples, tap water samples obtained in our laboratory were analyzed using this method. All water samples were spiked with different concentrations of $Fe^{3+}$ without any pretreatment, including the real samples and the N,S-CD probes. To examine the practical application of carbon dots, we analyzed the tap water samples obtained in our laboratory using spiked recovery. The collected water samples were treated with 0.22 μm filter membrane, and different concentrations of $Fe^{3+}$ were added to the real samples and the N,S-CD probes. Finally, the fluorescence emission spectra were recorded at the excitation wavelength of 360 nm for all samples.

## 3. Results and Discussion

### 3.1. Structure and Characterization of Carbon Dots

Using citric acid as a carbon source and thiourea as a nitrogen and sulfur source, high-QY fluorescent N,S-CDs were prepared by a one-step hydrothermal reaction. The morphology and particle size of the synthesized carbon dots were characterized by TEM. The TEM image in Figure 2a shows that the N,S-CDs are uniform in size, approximately

spherical, and have good dispersion. As can be seen from the inset of Figure 2a, the size distribution of the N,S-CDs is between 5 and 7 nm. As shown in Figure 2c, the XRD diagram exhibits a peak trend of the carbon dots, with a peak at $2\theta = 26.3°$, which corresponds to the (002) graphitic carbon diffraction curve and also corresponds to the lattice spacing of 0.21 nm (Figure 2b) [31,32]. Moreover, infrared spectroscopy (FTIR) was used to analyze the surface functional groups of the N,S-CDs to determine their water solubility. As shown in Figure 2d, which is from the perspective of the infrared spectrogram, the stretching vibration peak of C=S at 2064 cm$^{-1}$ increases with the decrease in the CA:NS ratio [33], and the stretching vibration peak of C=C [10–12] at 1412 cm$^{-1}$ also increases with the decrease in the ratio. Both functional groups are chromogenic groups, which affect the fluorescence emission of carbon dots. Moreover, the stretching vibration of O−H [15–17] at 3451 cm$^{-1}$, the stretching vibration of N–H [15–17] in –NH$_2$ at 3162 cm$^{-1}$, and the stretching vibration of C−N [22,26] at 1224 cm$^{-1}$ all provide the basis for the fluorescence properties of the auxochrome-based carbon dots. These are hydrophilic groups, which provide the carbon dots with good water solubility [21]. The change in the amide C=O's [27] bending vibration peak at 1715 cm$^{-1}$ gradually changed from a single peak to a double peak, which may be due to the structural changes caused by the changes in the amide reaction during product synthesis and may affect the fluorescence properties of the carbon dots. Combined with infrared spectroscopic analysis, it was shown that the N and S atoms were co-doped in the carbon dots in the form of functional groups, and functional groups such as hydroxyl, carboxyl, and amide groups existed on the surface of the N,S-CDs. These results are similar to those of many other reported CDs [18–27] and demonstrate that the prepared fluorescent products are CDs and have good water solubility.

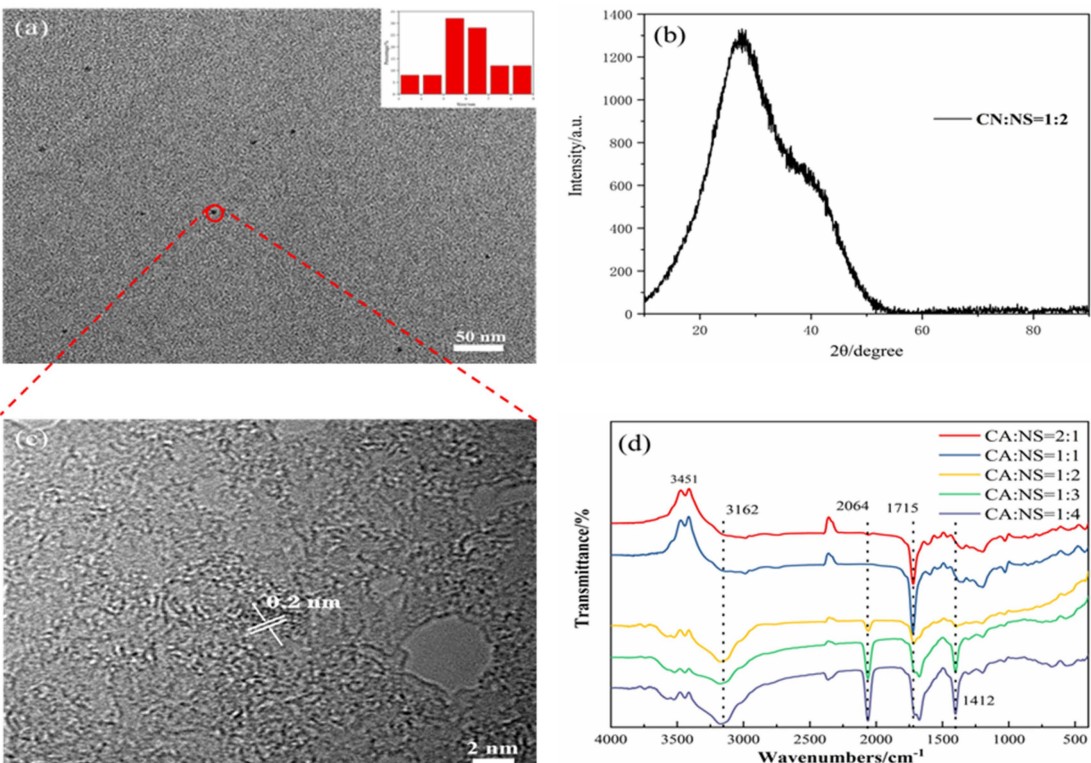

**Figure 2.** TEM and particle size distribution (**a**), XRD (**b**), lattice spacing (**c**), and IR spectrum (**d**) of N,S-CDs.

### 3.2. Optical Properties of Carbon Dots

The optical properties of the N,S-CDs were studied using UV-vis and fluorescence spectra. From the UV-Vis absorption spectrum in Figure 3b, the shoulder peak around 245 nm is due to the $\pi \rightarrow \pi^*$ energy transition of conjugated C=C hybridization, which

confirmed the formation of a ring reaction and the formation of heterocyclic structure in the amidation process [24]. The peak around 345 nm is related to the n→π* energy transition of C=O/C−O generated by the "surface band". This peak produces strong fluorescence emissions due to the electron holes (surface states) on the surface of the carbon dots that capture and release the excited-state energy [24].

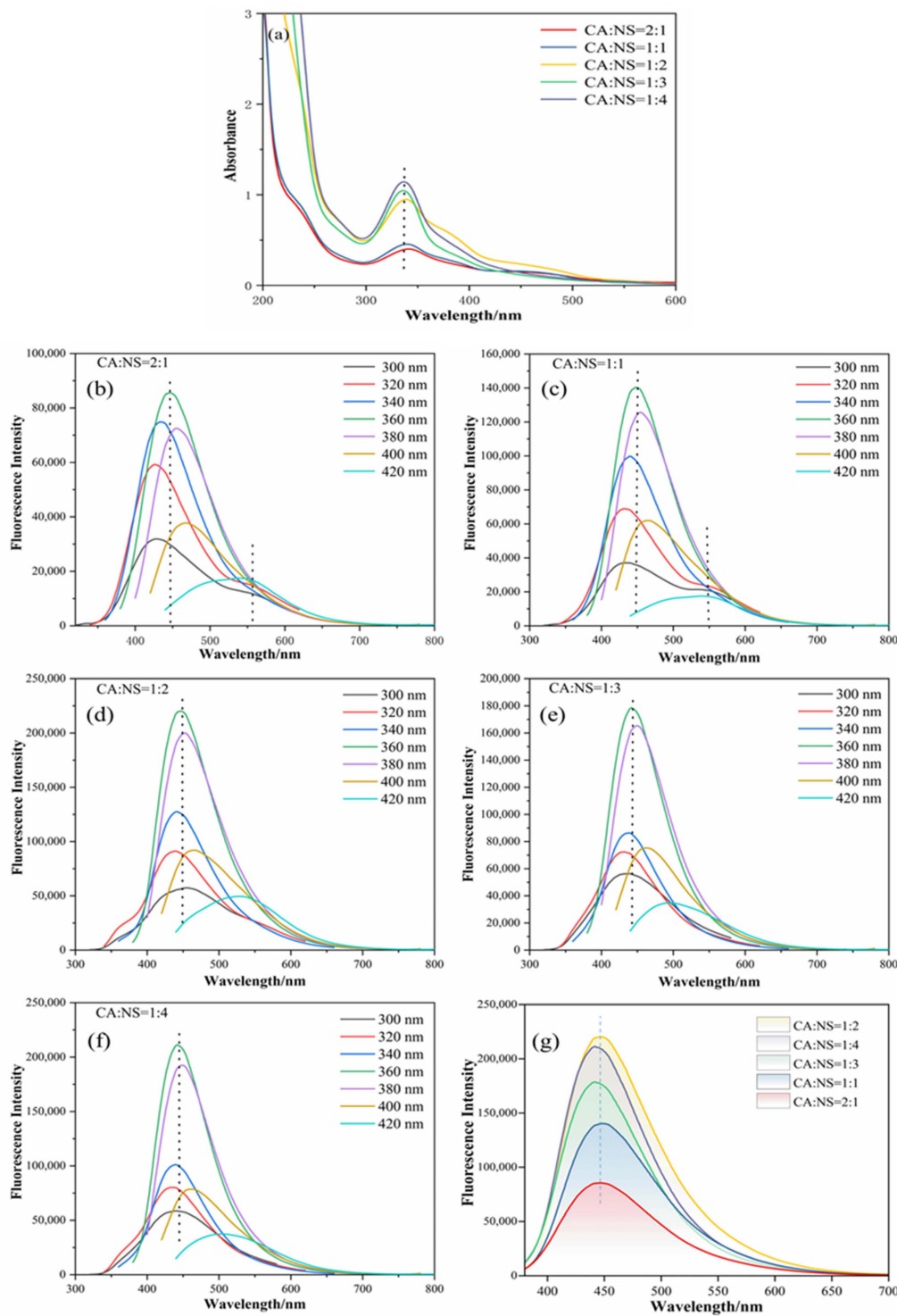

**Figure 3.** UV-Vis absorption spectra of different components of N,S-CDs (**a**), fluorescence emission spectra (**b**–**f**) at different excitation wavelengths from 300 to 420 nm, and emission spectra of each component at the optimal excitation wavelength of 360 nm (**g**).

In this experiment, the optical properties of the N,S-CDs were further characterized by measuring the fluorescence spectra of the N,S-CDs. The results of the fluorescence spectra of the N,S-CDs with different CA:NS ratios are shown in Figure 3b–f. It can be seen from the figure that with the change in the excitation wavelength from 300 nm to 400 nm, the fluorescence emission intensity first goes up and then comes down, and the characteristic emission peak of the N,S-CDs is obviously red-shifted. This indicates that different proportions of the N,S-CDs have obvious excitation light dependence. As can be seen in Figure 3b–f, different N,S-CDs all have the same optimal excitation wavelength, which is 360 nm. At the excitation wavelength of 360 nm, their corresponding emission peaks (445 nm, 447 nm, 450 nm, 445 nm, and 443 nm) are all in the blue emission range. Furthermore, in Figure 2b,c, there is a weak peak at 550 nm, which appears as a green fluorescence emission. However, at this time, the maximum emission peak of the N,S-CDs is around 450 nm, corresponding to the blue fluorescence emission, which overlays the green fluorescence emission, eventually leading to the blue fluorescence emission of the carbon dots. From the conclusions in these figures, it is inferred that the position of the starting emission peak is related to the excitation wavelength, and the emission peak has a red-shift phenomenon at different excitation wavelengths. These phenomena are induced by specific surface defects of the NS-CDs near the Fermi level, which can be illustrated by the IR spectra [26,34].

The optimal excitation wavelength of the N,S-CDs was determined to be 360 nm in Figure 3b–f. Figure 3g demonstrates the emission spectra of carbon dots with different doping ratios at the excitation wavelength of 360 nm. It was found that when the ratio of CA:NS was 1:2, the fluorescence intensity of the N,S-CDs was the highest. This is due to the unsaturated bond chromophore groups on the surface of the N,S-CDs, namely C=C and C=O, which lead the N,S-CDs to emit fluorescence under ultraviolet and visible light, and the fluorescence intensity is under the joint influence of the auxochrome group and the chromophore group. This resulted in the best fluorescence intensity at the ratio of 1:2.

### 3.3. Fluorescence Quantum Yield of Carbon Dots

It can be seen from the graph of the yield variation in each component in Figure 4 that with the decrease in the CA:NS ratio, the yield does not increase linearly, but presents an arched bridge trend, which first increased and then decreased with the increase in NS content. The maximum yield was achieved when the CA:NS ratio was 1:2, which was 53.80% (when the excitation wavelength was 360 nm), corresponding to the fluorescence emission intensity map in Figure 3g. The reason for this is that during the reaction of citric acid and thiourea, a large number of chromophore groups and auxochrome groups were generated on the surface of the carbon dots, and the surface oxidation degree was high, which affected the change in the fluorescence yield.

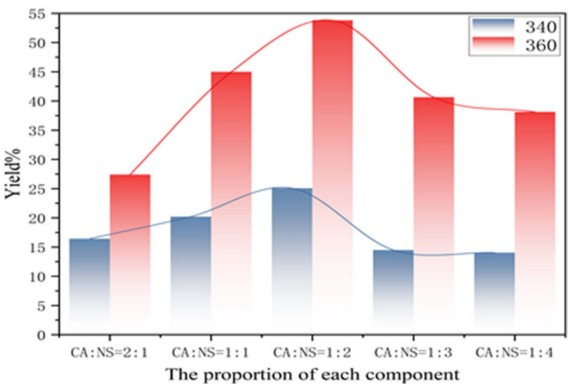

**Figure 4.** Variation in fluorescence yield in each component at excitation wavelengths of 340 and 360 nm.

### 3.4. Stability of Carbon Dots

For the purpose of exploring the stability of carbon dots, the carbon dots were first mixed with NaCl standard solution to make the final concentration of 0–2 mol/L. The influence of the NaCl standard solution concentration on the fluorescence intensity of the N,S-CDs is displayed in Figure 5a. F and $F_0$ represent the fluorescence emission intensities of the N,S-CDs in the presence and absence of metal ions, respectively. With the increase in the solution concentration, the fluorescence intensity of the N,S-CDs did not change much, which indicated that the N,S-CDs could exist stably in a high-concentration ionic environment. In addition, to further understand the stability of the carbon dots, the carbon dot solution was placed in different pH buffer solutions of pH 2–12, as shown in Figure 5b, which is the altered trend of the pH on the fluorescence intensity. From this, we can see that the fluorescence intensity of the N,S-CDs is about two times higher when the pH values are 6 and 7 than it is at pH 2 and pH 8–12. With the decrease in the pH value from 7 to 2 and the increase in acidity, the fluorescence intensity of the carbon dots weakened; as the pH increased from 7 to 12 and the alkalinity increased, the fluorescence intensity decreased significantly between 7 and 8 and then decreased slowly with the increase in the pH, demonstrating that the N,S-CDs have good fluorescence in neutral and weakly acidic environments. The luminescence intensity at the excitation wavelength of 360 nm was recorded as the photostability of the N,S-CDs under long-term storage at room temperature. As displayed in Figure 5c, with the passage of time at room temperature, the luminescence intensity of the carbon dots fluctuated up and down very little, demonstrating that the carbon dots have good photostability. In summary, the experiments show that the N,S-CDs have good stability.

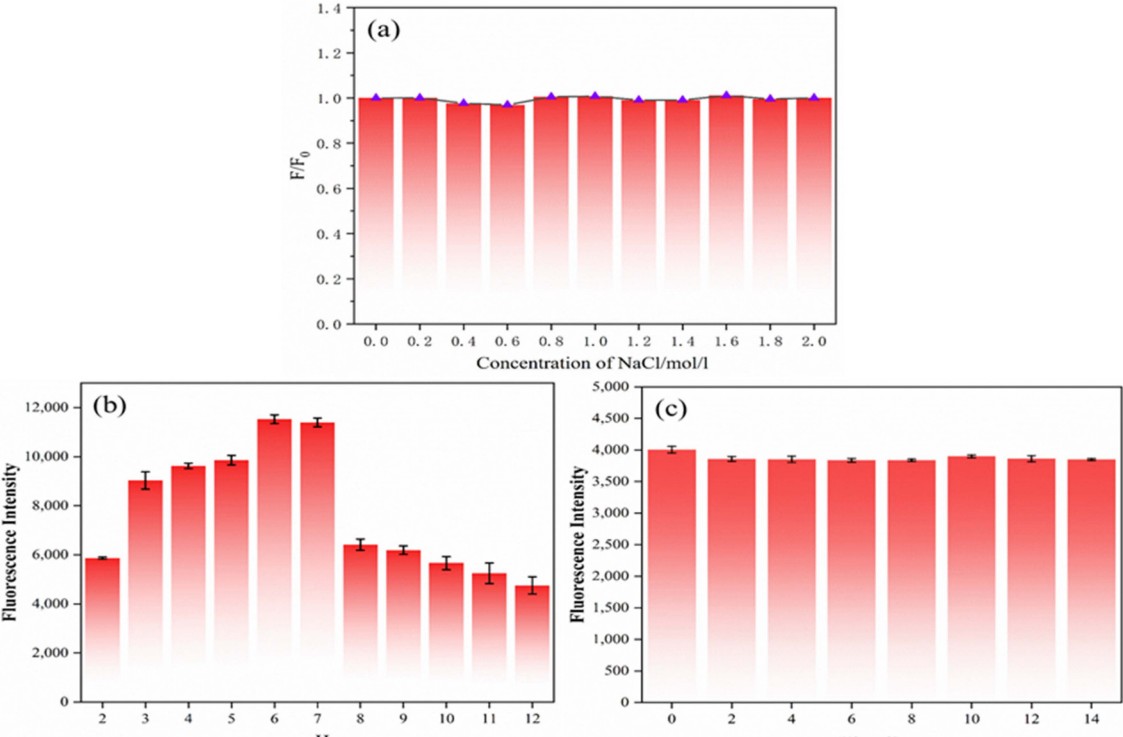

**Figure 5.** Effects of pH (**a**), NaCl solution concentration (**b**), and time (**c**) on fluorescence intensity of nitrogen–sulfur-doped carbon dots.

### 3.5. Metal Ion Selectivity

In order to evaluate the selective sensing ability and anti-interference ability of the N,S-CDs regarding metal ion selectivity, the fluorescence emission intensities of the N,S-CDs in the presence of different metal ions were investigated. Figure 6a illustrates the relative

luminescence emission intensity ($F/F_0$) of the metal ions at a concentration of 500 μM when the excitation wavelength is 360 nm. F and $F_0$ are the fluorescence emission intensities of the N,S-CDs in the presence and absence of metal ions. Different metal ions have different effects on the fluorescence intensity of the N,S-CDs. From the bar chart of the N,S-CDs mixed with metal ions (Figure 6a), it can be seen that the addition of $Fe^{3+}$ resulted in a significant decrease in the luminous intensity of the carbon dots and that the quenching was as high as 80%. Because the fluorescence intensity of other metal ions experienced no change or little change, the influence can be ignored. This shows that the prepared N,S-CDs have a stronger affinity for $Fe^{3+}$ and can have good selectivity to $Fe^{3+}$. Figure 6b indicates the effect of adding $Fe^{3+}$ on the relative fluorescence emission intensity ($F/F_0$) of the mixture system of metal ions and N,S-CDs. It can be seen from the bar chart (Figure 6b) that the addition of $Fe^{3+}$ into the mixed system can still reduce the fluorescence intensity of the N,S-CDs to the greatest extent and that other metal ions cannot affect the quenching effect of $Fe^{3+}$ on the N,S-CDs. This proves that the prepared N,S-CDs have good ionic anti-interference ability.

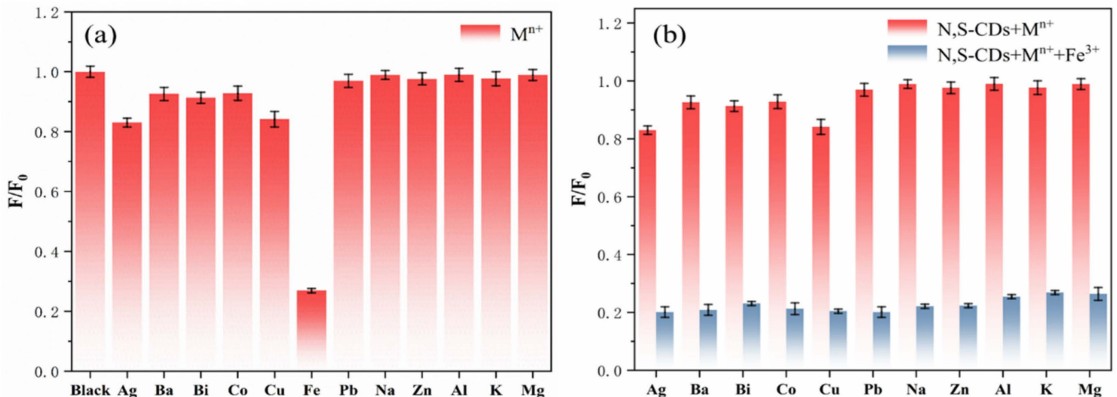

**Figure 6.** Selectivity (**a**) and anti-interference ability (**b**) of nitrogen–sulfur-doped carbon dots for different ions.

### 3.6. Linear Analysis of Carbon Dots on $Fe^{3+}$

The linear analysis of $Fe^{3+}$ can be continued by using the stability of the N,S-CDs mentioned above. As displayed in Figure 7a,b, the relative fluorescence intensity changes and fluorescence intensity diagrams of the nitrogen–sulfur-doped carbon dots with various concentrations of $Fe^{3+}$ (0, 10, 20, 30, 40, 50, 60, 70, 80, 90, 100, 200, 400, 500, and 600 μM) are described. This can clearly show that when the concentration of $Fe^{3+}$ increases, the intensity of the N,S-CDs also decreases. As displayed in Figure 7c, in the range of 0–100 μM, the relative fluorescence intensity ($F/F_0$) displayed a good linear relationship, with an $Fe^{3+}$ concentration ($R^2 = 0.9965$). According to the formula $D = 3\sigma/m$ [19,23,26–28], the detection limit (D) was obtained as 0.20 μM (about 0.11 mg/L). Compared to the detection limits in the literature (Table 1), the carbon dots prepared in this paper have higher quantum yields and lower detection limits, which can be used for actual sample detection.

**Table 1.** Performance comparison of the different detection ranges and detection limits of carbon dots in the literature.

| Sensing Platform | Linear Range (μM) | Quantum Yield | LOD (μM) | Reference |
|---|---|---|---|---|
| Cu-NCs | 1–100 | 8.6% | 0.3 | [28] |
| N-CDs | 0.002–8 | 30.2% | 0.138 | [19] |
| C-dots | 10–100 | 28.6% | 0.398 | [27] |
| N-CDs | 8–80 | No report | 3.8 | [23] |
| N-CDs | 5–500 | No report | 0.720 | [26] |
| N,S-CDs | 0–100 | 53.80% | 0.2 | This Work |

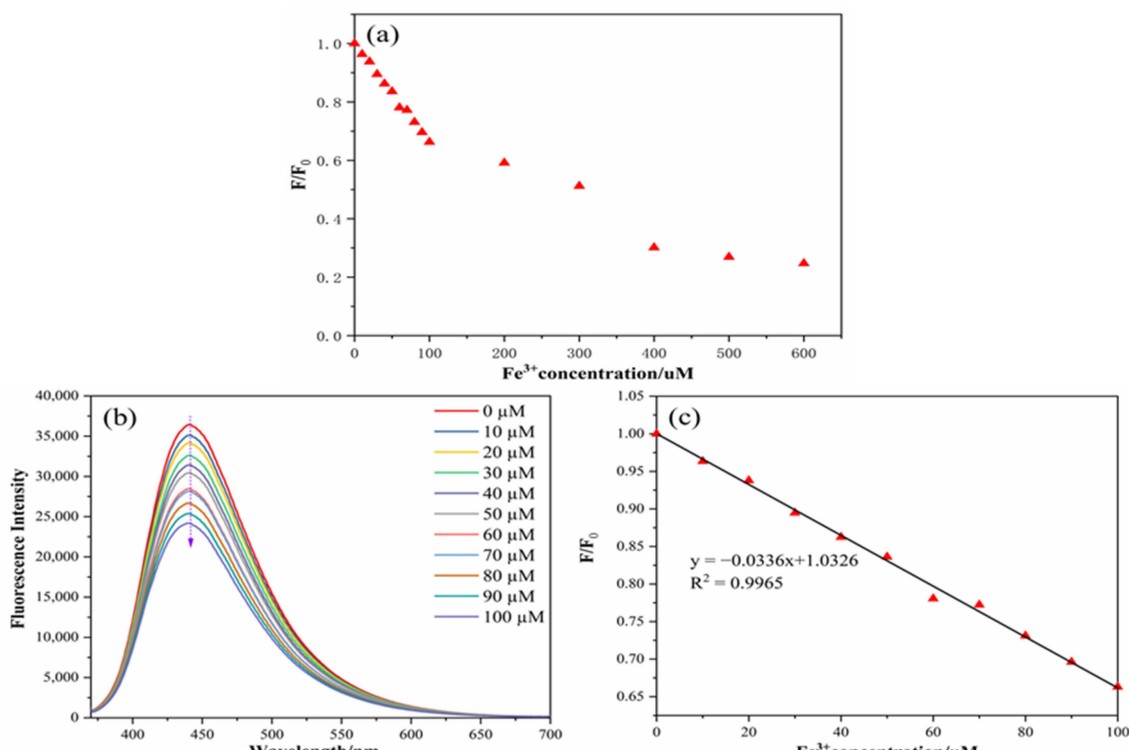

**Figure 7.** Relative fluorescence intensity changes in N,S-CDs under different concentrations of $Fe^{3+}$ (**a**), fluorescence intensity spectra of N,S-CDs under different concentrations of $Fe^{3+}$ (**b**), and linear relationship between $Fe^{3+}$ concentration and fluorescence intensity of the N,S-CDs (**c**).

Finally, the quenching mechanism of $Fe^{3+}$ on the N,S-CDs was briefly explored through the infrared and ultraviolet absorption spectra of the N,S-CDs and N,S-CDs+$Fe^{3+}$. Figure 8a displays the UV absorption spectrum in the mixed system of N,S-CDs+$Fe^{3+}$, and an absorption curve that is different from that of single N,S-CDs appeared. The curve has absorption peaks centered at 300 nm and 365 nm, and the peaks are higher than those in the N,S-CDs alone. In the infrared spectrum as displayed in Figure 8b, it can be observed that the stretching vibration peak of O–H of the hydroxyl group at 3451 $cm^{-1}$ in the mixed system of N,S-CDs+$Fe^{3+}$ is significantly weakened compared to that of the N,S-CDs alone, while the other peaks remain unchanged. This explains that when $Fe^{3+}$ is added, the hydroxyl functional groups on the surface of the N,S-CDs are complexed with them to form ground-state complexes, which are attached to the surfaces of the carbon dots, affecting the luminescence properties of the N,S-CDs and resulting in the quenching effect of $Fe^{3+}$.

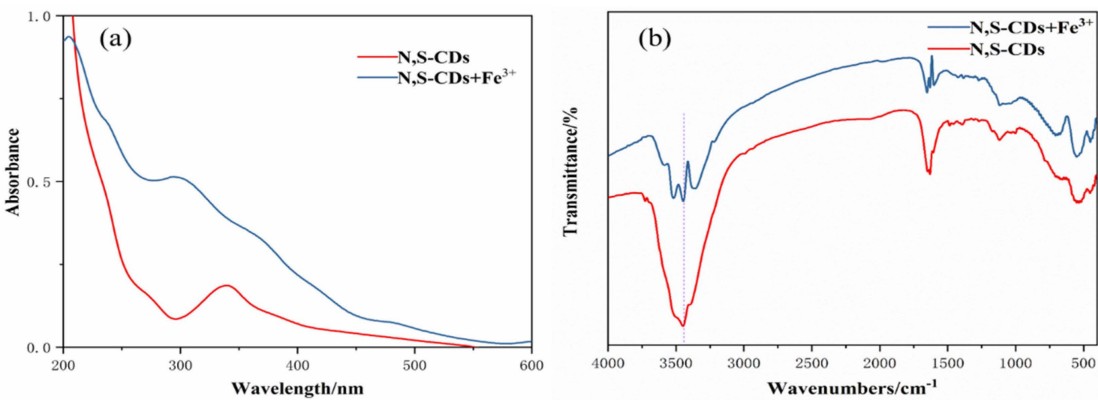

**Figure 8.** UV spectra (**a**) and infrared spectra (**b**) of N,S-CDs+$Fe^{3+}$ system and N,S-CDs.

*3.7. Real Sample Analysis*

To further prove the effectiveness of N,S-CDs in detecting $Fe^{3+}$, three groups of $Fe^{3+}$ with different concentrations were injected into a solution mixed with carbon dots and local tap water, and the data were recorded three times. The conclusion is displayed in Table 2, where the recovery rate of $Fe^{3+}$ in tap water is 97.3%–110.1%, and the relative standard deviation is not more than 4%. The conclusion indicates that the N,S-CDs prepared in this paper had sufficient reliability and sensitivity and that the iron content indicated in the national sanitary standard for drinking water (GB 5749-2006) was 0.3 mg/L (5 μM) [8], which was higher than the detection limit in this paper, proving that the N,S-CDs could be used to measure the iron ion concentration in drinking water.

**Table 2.** The recovery rate of standard addition of $Fe^{3+}$ in tap water.

| Number | Added (μM) | Found (μM) | Recover (%) | RSD (%, n = 3) |
|--------|-----------|-----------|-------------|----------------|
| 1 | 1 | 0.99 | 97.3 | 3.28 |
| 2 | 5 | 5.56 | 110.1 | 1.91 |
| 3 | 10 | 10.23 | 101.8 | 1.05 |

## 4. Conclusions

In this study, co-doped nitrogen–sulfur carbon dots (N,S-CDs) were synthesized hydrothermally by employing thiourea and citric acid as precursors. By changing the ratio of citric acid and thiourea, blue fluorescent N,S-CDs with a quantum yield of 53.80% were synthesized. In weakly acidic, neutral, and high-concentration ionic environments, the N,S-CDs have a highly selective quenching response to $Fe^{3+}$. When the concentration is 0–100 μM, the fluorescence intensity of the N,S-CDs has a linear relationship with the concentration of $Fe^{3+}$ ($R^2$ = 0.9965), and the measured detection limit (D = 3σ/m) is 0.2 μM. The results reported in this paper demonstrates the prospective potential application of carbon dots in the field of environmental analysis and monitoring.

**Author Contributions:** Project administration, Z.L. (Zhengxin Li); Supervision, W.H.; Writing—original draft, H.Z.; Writing—review & editing, Y.R.; Help experiment, Z.L. (Zheng Li). All authors have read and agreed to the published version of the manuscript.

**Funding:** This work was financially supported by the Innovative Funds Plan of Henan University of Technology (No. 2021ZKCJ06) and supported by the Project of Henan Science and Technology Research Program in 2022 (No.222102230040) and the Cultivation Programme for Young Backbone Teachers in Henan University of Technology (No. 21420117).

**Institutional Review Board Statement:** Not applicable.

**Informed Consent Statement:** Not applicable.

**Data Availability Statement:** The supporting data are available in Henan University of Technology and can be availed upon reasonable request. The data sets generated during this study are available from the corresponding author upon reasonable request.

**Conflicts of Interest:** The authors declare no conflict of interest.

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
