# Peer review of "Selective Detection of Fe3+ by Nitrogen–Sulfur-Doped Carbon Dots Using Thiourea and Citric Acid"

_coatings, doi:10.3390/coatings12081042_

Round 1

Reviewer 1 Report

I have gone through the manuscript entitled " Selective detection of Fe3+ by nitrogen-sulfur doped carbon dots using thiourea and citric acid".

I am sure that the study deserves the attention of readers and can be published; however, some major concerns need to be addressed before accepting the paper for publication to improve the readability and clarity of the manuscript:

1.     Line 40 – “widely”

2.     Lines 75-76 It seems to be an incomplete sentence…

3.     Lines 94-95 The text should be formatted in “italic”?

4.     Line 108: In what form were these NS-CDs obtained? In solution,  powder? It must be specified!

5.     Lines 122-126: The authors are asked to revise this paragraph. What does "diluent" mean?

6.     Line 129: it must be clarified how this dilution was obtained

7.     Line 138: what is “sTo”? Please, revise…

8.     Line 140: should be “µm”

9.     Line 155: …from…

10.  Line 156: What is "infrared structure"?

11.  Lines 157-170:

- I assume that “CA” is citric acid. The abbreviation must be added to the text after the first mention of the abbreviated word;

- “the stretching vibration peak of C=S at 2064 cm-1 increases with the increase of the CA:NS ratio…  Isn't it a decrease? The intensity of the peak increases with the concentration of NS, so with the decrease of the ratio CA: NS. This aspect must be clarified wherever it appears in the text (lines: 158, 228, 292, etc);

- How was the FTIR analysis performed? Transmission or ATR, solid or liquid samples? These aspects should be included in the materials and methods section;

- Bibliographic references must be added to the identified FTIR bands;

- The positions of the FTIR bands must be written in the same format! Use, for example, “3162 cm-1” instead of “3162cm-1” ...

12. Line 163: What do "other characteristics" mean? A reference must be added…

13. Lines 167-170: “Combined with the analysis in the infrared spectrum, it is shown that N and S atoms are co-doped in carbon dots in the form of functional groups, and functional groups such as hydroxyl, carboxyl and amide exist on the surface of N,S-CDs, which proves the good water solubility of carbon dots “. Please, revise…

14. Line 172: Why is the structure of peaks different depending on the concentration of NS?

15. Line 182: Were these "cavities" observed experimentally in this study? Otherwise, it is necessary to add a reference.

16. Lines 188-213: I suggest a shortening of the paragraph. The behavior is almost similar, being observed the same tendency to change the maximum fluorescence with the excitation wavelength…

17. Line 216: add a reference

18. Lines 232-235: Why is the number of these groups higher in CA: NS = 1: 2 than in the other ratios? How was the oxidation of the surface evaluated?

19. Line 239: In figure 4a what does F / F0 represent?

Also, the experimental aspects related to the stability of NS-CDs should be introduced in the Materials and Methods section;

20. Line 251: Compared to what is this change in fluorescence intensity “little”? From Fig. 4b it is observed that at pH 6 and 7, the intensity values are almost double compared to those at pH 2, 8-12 ...

21. Line 294: A reference for the equation of “detection limit (D)” should be added

22. Lines 321-324 I suggest a revision of this paragraph. From my point of view, the term "the way" does not seem to be appropriate in this expression.

Reviewer 2 Report

The research work presented by Ying Ren's group in the research article (No. coatings-1779080) was submitted for the publication in 'coatings' with the title 'Selective detection of Fe3+ by nitrogen-sulfur doped carbon dots using thiourea and citric acid' describes the synthesis of nitrogen-sulfur-doped carbon dots (N,S-CDs) by the hydrothermal technique using the varying ratios of thiourea (nitrogen source) and citric acid. With the adjustment of the quantities of citric acid and thiourea along with the addition of dry ethanol, the authors were able to prepare the fluorescent doped carbon dots with quantum yield up to 53.80%. The fabricated materials were systematically characterized using TEM, XRD, and FT-IR techniques. The resultant materials with suitable functional groups showed good chelation ability to Fe+3 and showed high selectivity using Fe+3 concentrations between 0 to 100 μM. Further, a linear relation between the fluorescence intensity of N,S-CDs and Fe+3 (R2=0.9965) was identified, with a detection limit (D=3ơ/m) measured as 0.2 μM.

The manuscript is well written and well organized. The research proposed in this article deals with the fabrication of N,S-CDs having good optical properties is highly helpful in the Fe+3 detection related to the environmental monitoring since high amounts of iron ions are toxic to human health and the environment. Hence this article can be accepted for publication in coatings after addressing the following minor concerns.

1. It was used 'For the sake of' in several places in the manuscript, could it be modified?

2. Some information regarding the urgency of removal/detection of metal ions should be included in the introduction. In this connection the useful articles are Envionmental Chemistry Letters, 2019, 17, 1495–1521; Membranes 2021, 11, 792; Chemical Communications 2021, 57, 7215-7231; Journal of Environmental Chemical Engineering, 2021, 9, 106553. The citation of these articles along with others also improves the reference section.

3. The references should be formatted according to the format of coatings.

Round 2

Reviewer 1 Report

The paper could be published in present form.